# Start-Up Electroosmotic Flow of Multi-Layer Immiscible Maxwell Fluids in a Slit Microchannel

**DOI:** 10.3390/mi11080757

**Published:** 2020-08-05

**Authors:** Juan Escandón, David Torres, Clara Hernández, René Vargas

**Affiliations:** 1Instituto Politécnico Nacional, SEPI-ESIME Azcapotzalco, Departamento de Termofluidos, Av. de las Granjas No. 682, Col. Santa Catarina, Alcaldía Azcapotzalco, Ciudad de México 02250, Mexico; dtorresc1600@alumno.ipn.mx (D.T.); reneosvargas@yahoo.com.mx (R.V.); 2Universidad Tecnológica de México -UNITEC MÉXICO- Campus Marina-Cuitláhuac, Ciudad de México 02870, Mexico; clherrob@mail.unitec.mx

**Keywords:** electroosmotic flow, microchannel, immiscible fluids, electrostatic effects, interfacial phenomena, Maxwell fluids, parallel flows

## Abstract

In this investigation, the transient electroosmotic flow of multi-layer immiscible viscoelastic fluids in a slit microchannel is studied. Through an appropriate combination of the momentum equation with the rheological model for Maxwell fluids, an hyperbolic partial differential equation is obtained and semi-analytically solved by using the Laplace transform method to describe the velocity field. In the solution process, different electrostatic conditions and electro-viscous stresses have to be considered in the liquid-liquid interfaces due to the transported fluids content buffer solutions based on symmetrical electrolytes. By adopting a dimensionless mathematical model for the governing and constitutive equations, certain dimensionless parameters that control the start-up of electroosmotic flow appear, as the viscosity ratios, dielectric permittivity ratios, the density ratios, the relaxation times, the electrokinetic parameters and the potential differences. In the results, it is shown that the velocity exhibits an oscillatory behavior in the transient regime as a consequence of the competition between the viscous and elastic forces; also, the flow field is affected by the electrostatic conditions at the liquid-liquid interfaces, producing steep velocity gradients, and finally, the time to reach the steady-state is strongly dependent on the relaxation times, viscosity ratios and the number of fluid layers.

## 1. Introduction

Microfluidics is a term that is used in fields of science with miniaturized systems where fluids are used as key components of control and sensing [1]. The handling of small sample volumes, scalability, integration of multiple functions and fields, low operating costs, low energy consumption, and so forth are some of the already known advantages of systems miniaturization; however, an inherent problem in these small devices is in the manufacture of moving parts for the manipulation of fluids or samples. That is why techniques based on electrokinetic effects arise that do not need moving parts [2]. The microsystem known as a laboratory on a chip, with size from millimeters to centimeters, facilitates the implementation of many laboratory tasks, which include sample preparation, mixing, separation, manipulation, control, detection, and analysis [3]. Applications cover the areas of mechanics, biology, chemistry, and medicine, seeking to improve technologies to preserve human health and improve the quality of life [4].

In this context, and to cover the different applications above mentioned, the electroosmotic flow has emerged as an electrokinetic phenomenon to transport fluids in microsystems. The electroosmosis represents the movement, due to an applied electric field, of an electrolyte solution relative to a stationary charged surface [5]. This transport method has been theoretically studied since many years ago in small channels with Newtonian fluids, as the early works carried out by Burgreen and Nakache [6] and Rice and Whitehead [7]. Since then and until now, the scientific community has continued studies on the electroosmotic flow behavior using Newtonian [8,9,10,11,12] and non-Newtonian fluids [13,14,15,16,17,18,19], addressing emerging issues regarding the rheology of the fluids and ionic concentrations, channel geometry, wall zeta potentials effects, boundary slip effects, among other topics, and their implications on the flow characteristics.

Research progress on electroosmotic flows with single-phase fluids is broad, as shown through the references cited in the previous paragraph and the references contained within them in turn. However, several microdevices applications use parallel multiphase flows for continuous chemical processing in analyses and synthesis [20,21], realizing specific operations as mixing and reaction [22], phase confluence and separation [23], solvent extraction [24,25], liquid-liquid extraction [26], purification [27] and synthesis of polymer membranes [28]. These parallel flows are a kind of flow pattern generated by using flow-focusing techniques in microsystems [29,30].

Following this direction, the scientific community exploits the benefit of electrokinetic phenomena by not including moving parts in microdevices, developing electrokinetic flow-focusing techniques. Hence, Pan et al. [31] perform an experimental investigation in microfluidic chips for mixing enhancement through the electrokinetic flow focusing and valveless switching of multiple samples flows. Meanwhile, Jiang et al. [32] propose a new microfluidic method to transport samples between sheath streams; here, the sheath streams for flow-focusing are generated by electroosmotic effects. In another work, Li et al. [33] present a theoretical and experimental investigation on flow-focusing with valveless switching, using the coupled effect of hydrodynamics and electroosmosis; in this work, the applied technique for switching a nonconducting sample stream or droplets, use two sheath streams of conducting fluids in a microchannel under electroosmotic effects. In this sense, Jia et al. [34] carried out a continuous-flow focusing study for collecting microparticles using induced-charge electroosmosis in a microfluidic device.

Therefore, for about two decades, the research to understand the physical mechanisms for moving parallel flows of immiscible fluids under electrokinetic effects has considered the transport of two layers [35,36,37,38,39,40,41,42] and three layers [43]. In these investigations, the arrangement of fluids considers that one fluid layer is non-conducting and the other(s) fluid(s) layer(s) if, being the electrolytic fluid(s) which is(are) under electroosmotic effects. Consequently, interfacial phenomena include the formation of an electrical double layer both in solid-liquid interfaces and in liquid-liquid interfaces. Regarding the liquid-liquid interfaces, the hydrodynamic and electrostatic boundary conditions are established in a relatively simple form with a specified zeta potential and the partially or null employment of Maxwell electric stress. However, other studies about the flow of two immiscible parallel fluids, consider that both fluids are conductive (i.e., fluids based in electrolytic solutions), increasing the complexity of the electrostatic boundary conditions in the liquid-liquid interface through a potential difference and the Gauss’s law for the electrical displacement, together with the hydrodynamic boundary conditions via the combination of viscous and electric Maxwell stresses [44,45,46,47]. In addition, to cover the different flow-focusing applications in microdevices, the study of parallel flows under electrokinetic effects also has been extended to multi-layer systems [48,49,50].

To complement the research of Liu et al. [36], Li et al. [48], Afonso et al. [38], Huang et al. [39], Jian et al. [46], Matías et al. [42], Escandón et al. [49] about parallel flows with non-Newtonian fluids under electroosmotic effects, and considering that many of handled fluids in microsystems have complex rheological behavior, the present investigation aims to realize a parametric study on the start-up of the electroosmotic flow of multi-layer immiscible Maxwell fluids in a microchannel. The semi-analytical solution for the velocity field that is based on the Laplace transform method, obtains the description of new liquid-liquid interfacial phenomena as well as combined effects from the physical, rheological and electrical properties of fluids, over the velocity profiles and on the tracking of the velocity magnitude in the time. Hence, this model will be able to analyze many cause-effect relationships and it should serve as a guide for starting an experimental set-up in microdevices which require high spatio-temporal precision to carry out different clinical, chemical and biological analyses.

## 2. Mathematical Modeling

### 2.1. Physical Model Description

In this work, the transient electroosmotic flow of multi-layer immiscible fluids in a slit microchannel is analyzed. The physical model of the flow phenomenon studied here is on a Cartesian coordinate system (x,y), as is shown in Figure 1. The conduit is integrated by two parallel flat plates separated by a distance of 2H and filled by fluid layers, in which each one is composed of a mixture of symmetrical electrolytes with solutes that exhibit viscoelastic behavior. In the sketch, each liquid-liquid interface is placed in a $y_{n}$ position; where, the subscript n=1,2,3,...,i represents the number of the fluid layer, and *i* is the fluid layer in contact with the upper microchannel wall. Due to the fluids are immiscible and electrically conductive, in addition to the fact that the interfaces between them are polarizable and impermeable to charged particles, in this region, a double electrical layer appears presenting electrostatic properties through a potential difference $\Delta \psi_{n}$. The microchannel walls are also polarizable and acquire a surface electric charge represented by the zeta potential $\zeta_{w}$, also promoting the formation of an electric double layer in the solid-liquid interfaces. The fluids movement is due to the ends of the conduit are subject to an electric potential generated by a pair of electrodes that gives rise to a uniform electric field $E_{x}$ inducing electroosmotic effects on the electric double layers described before.

### 2.2. Governing and Constitutive Equations

The flow field of multi-layer immiscible fluids is governed by the Poisson equation for the electric potential distribution on the microchannel
(1)$$\nabla^{2}\Phi_{n}=-\frac{\rho_{e,n}}{\epsilon_{n}},$$
where $\Phi_{n}$ is the electric potential, $\rho_{e,n}$ is the volumetric free charge density and $\epsilon_{n}$ is the dielectric permittivity. Also, with the continuity equation for incompressible fluids as
(2)$$\nabla \cdot v_{n}=0,$$
where v is the velocity vector. And the Cauchy momentum governing equation
(3)$$\rho_{n}\frac{Dv_{n}}{Dt}=-\nabla p+\nabla \cdot \tau_{n}+\rho_{n}g+\rho_{e,n}E,$$
where $\rho_{n}$ is the fluid density, *t* is the time, *p* is the pressure, $\tau_{n}$ is the stress tensor, g is the gravitational acceleration vector and E is the electric field vector. The shear stress tensor is related with the Maxwell rheological model as [51,52]
(4)$$\tau_{n}+\lambda_{n}\frac{\partial \tau_{n}}{\partial t}=-\eta_{0,n}{\dot{\gamma }}_{n},$$
where $\lambda_{n}$ is the relaxation time, $\eta_{0,n}$ is the zero-shear-rate viscosity and $\dot{\gamma_{n}}=\left(\nabla v_{n}\right)+{\left(\nabla v_{n}\right)}^{T}$ is the rate-of-strain-tensor.

### 2.3. Simplified Mathematical Model

The general governing equations given in the previous section can be simplified taking into account the following assumptions:Constant physical properties and independent of the local electric field, ion concentration, and temperature [36,53].Fully-developed flow [35].Impermeable interfaces between the fluids and ideally polarizable to electric charges [5,54,55,56]. The electric double layers at the liquid-liquid interfaces are composed of two diffuse charge layers separated by a central compact layer, characterized by a potential difference drop due to the orientation of the solvent molecules; also, the continuity of electrical displacements on both sides of the central inner layer, in absence of ions in the inner layer is considered [5,55,56,57,58].Flat interfaces between the fluids [44,59,60,61]. This is assumed when considering that there is: (i) creeping flow for low Reynolds numbers, being $Re_{n}\left(=\rho_{n}Hu_{HS}/\eta_{0,n}\right)\ll 1$, resulting in parallel flows with laminar fluid interfaces [43], and (ii) uniform zeta potentials along the microchannel. Here, the characteristic velocity of flow is the well-known Helmholtz-Smoluchowski velocity defined by $u_{HS}=-\epsilon_{\mathrm{ref}}\zeta_{w}E_{x}/\eta_{\mathrm{ref}}$, where the subscript “ref” indicates physical properties referred to electrolytes in aqueous solutions at 298.15 K (25 ${}^{\circ }$C) [5,62].The gravitational forces are neglected [60].Long microchannel neglecting any end effects; hence, the electric potential $\Phi_{n}$, is the algebraic sum of the potential due to the electric double layer, $\psi_{n}$, and the potential due to the imposed electric field, ϕ, as [5]:
(5)$$\Phi_{n}\left(x,y\right)=\psi_{n}\left(y\right)+\phi \left(x\right),$$
where $\phi \left(x\right)=\phi_{0}-xE_{x}$; being $\phi_{0}$ the electric potential at the inlet of the microchannel at x=0 and $E_{x}$ is the external electric field independent of the position and constant along the axial direction.The local distribution of the free charges, that is, ions, is governed by the electrical potential into the electric double layer, $\psi_{n}$, through the Boltzmann distribution as [5]
(6)$$\rho_{e,n}=-2z_{n}en_{0,n}\sinh\left(\frac{z_{n}e\psi_{n}}{k_{B}T_{n}}\right),$$
where $z_{n}$ is the valence of electrolyte, *e* is the electron charge, $n_{0,n}$ is the ionic number concentration in the bulk solution, $k_{B}$ is the Boltzmann constant and $T_{n}$ is the fluid temperature.The Debye-Hückel approximation for small interfacial potentials at the solid-liquid [5,63] and liquid-liquid interfaces [45,46] is used. This approximation can be valid for values up to 50 mV [5,7,9].There is no imposed pressure gradient on microchannel.The electric double layers do not overlap.Any physical or chemical modification on the wall surfaces to cause hydrophobic interactions at the solid-liquid interfaces [47,64] is negligible.

As result, Equations (Equation 1)–(Equation 4), can be rewritten for a unidirectional flow as follows, leaving the Poisson-Boltzmann equation
(7)$$\frac{d^{2}\psi_{n}\left(y\right)}{dy^{2}}=\kappa_{n}^{2}\psi_{n}\left(y\right),$$
the momentum equation
(8)$$\rho_{n}\frac{\partial u_{n}\left(y,t\right)}{\partial t}=-\frac{\partial \tau_{xy,n}\left(y,t\right)}{\partial y}-\epsilon_{n}\kappa_{n}^{2}E_{x}\psi_{n}\left(y\right)$$
and respectively the Maxwell’s constitutive rheological equation
(9)$$\tau_{xy,n}\left(y,t\right)+\lambda_{n}\frac{\partial \tau_{xy,n}\left(y,t\right)}{\partial t}=-\eta_{0,n}\frac{\partial u_{n}\left(y,t\right)}{dy},$$
where $\kappa_{n}^{2}=2z_{n}^{2}e^{2}n_{0,n}/\epsilon_{n}k_{B}T_{n}$ is the Debye-Hückel parameter related to the Debye length $\kappa_{n}^{-1}\;=\;{\left(\epsilon_{n}k_{B}T_{n}/2z_{n}^{2}e^{2}n_{0,n}\right)}^{1/2}$ [5], $u_{n}$ is the velocity on the x−direction and $\tau_{xy,n}$ is the shear stress.

However, to get a momentum equation only in terms of velocity, Equation (Equation 9) is derived with respect to the transverse coordinate, producing
(10)$$\frac{\partial \tau_{xy,n}\left(y,t\right)}{\partial y}=-\lambda_{n}\frac{\partial^{2}\tau_{xy,n}\left(y,t\right)}{\partial t\partial y}-\eta_{0,n}\frac{\partial^{2}u_{n}\left(y,t\right)}{\partial y^{2}},$$
which is replaced into Equation (Equation 8) obtaining
(11)$$\rho_{n}\frac{\partial u_{n}\left(y,t\right)}{\partial t}=\lambda_{n}\frac{\partial^{2}\tau_{xy,n}\left(y,t\right)}{\partial t\partial y}+\eta_{0,n}\frac{\partial^{2}u_{n}\left(y,t\right)}{\partial y^{2}}-\epsilon_{n}\kappa_{n}^{2}E_{x}\psi_{n}\left(y\right).$$

On the other hand, Equation (Equation 8) is derived with respect to time, yielding
(12)$$\rho_{n}\frac{\partial^{2}u_{n}\left(y,t\right)}{\partial t^{2}}=-\frac{\partial^{2}\tau_{xy,n}\left(y,t\right)}{\partial t\partial y}.$$

The previous result is replaced into Equation (Equation 11), obtaining a momentum equation of hyperbolic type for the *n* fluids in terms of the axial velocity as follows
(13)$$\rho_{n}\lambda_{n}\frac{\partial^{2}u_{n}\left(y,t\right)}{\partial t^{2}}+\rho_{n}\frac{\partial u_{n}\left(y,t\right)}{\partial t}=\eta_{0,n}\frac{\partial^{2}u_{n}\left(y,t\right)}{\partial y^{2}}-\epsilon_{n}\kappa_{n}^{2}E_{x}\psi_{n}\left(y\right).$$

To solve the governing equations given in Equations (Equation 7) and (Equation 13), the following boundary conditions in t>0 for the electric potential and velocity are considered. At the bottom wall of microchannel for the fluid layer n=1, the boundary conditions at y=0 are a specified zeta potential and the no-slip boundary condition respectively as
(14)$$\psi_{1}\left(y=0\right)=\zeta_{w},\;u_{1}\left(y=0,t\right)=0.$$

In the case of each liquid-liquid interface at $y=y_{n=1,2,3,...,i-1}$, the boundary conditions that are considered are a potential difference, the Gauss’s law for the electrical displacement, a velocity continuity, and a stresses balance that include the Maxwell stresses and viscous shear stresses (also called electro-viscous stresses balance), respectively as follows
(15)$$\psi_{n+1}\left(y\right)-\psi_{n}\left(y\right)=\Delta \psi_{n},$$
(16)$$\epsilon_{n+1}\frac{d\psi_{n+1}\left(y\right)}{dy}=\epsilon_{n}\frac{d\psi_{n}\left(y\right)}{dy},$$
(17)$$u_{n}\left(y,t\right)=u_{n+1}\left(y,t\right)$$
and
(18)$$\tau_{xy,n}\left(y,t\right)+\tau_{e,n}\left(y\right)=\tau_{xy,n+1}\left(y,t\right)+\tau_{e,n+1}\left(y\right),$$
where the Maxwell shear stress is
(19)$$\tau_{e,n}\left(y\right)=-\epsilon_{n}E_{x}\frac{d\psi_{n}\left(y\right)}{dy}.$$

Additionally, at the top wall of microchannel for the fluid layer n=i, the boundary conditions at y=2H, are a specified zeta potential and the no-slip boundary condition respectively as
(20)$$\psi_{i}\left(y=2H\right)=\zeta_{w},\;u_{i}\left(y=2H,t\right)=0.$$

Finally, the initial conditions to solve the momentum equation, Equation (Equation 13), for the entire geometric domain 0≤y≤2H, that is, for all fluid layers are
(21)$$u_{n}\left(y,t=0\right)=0,\;\tau_{xy,n}\left(y,t=0\right)=0,\;{\left.\frac{\partial u_{n}}{\partial t}\right|}_{y,t=0}=0.$$

### 2.4. Dimensionless Mathematical Model

The mathematical model is normalized with the following dimensionless variables
(22)$$\bar{t}=\frac{\eta_{\mathrm{ref}}t}{\rho_{\mathrm{ref}}H^{2}},\;\bar{y}=\frac{y}{H},\;{\bar{\psi }}_{n}=\frac{\psi_{n}}{\zeta_{w}},\;{\bar{u}}_{n}=\frac{u_{n}}{u_{HS}},\;{\bar{\tau }}_{xy,n}=\frac{H\tau_{xy,n}}{\eta_{\mathrm{ref}}u_{HS}}.$$

Therefore, by replacing Equation (Equation 22) in Equations (Equation 7), (Equation 9) and (Equation 13), the dimensionless version of the governing and constitutive equations of Poisson-Boltzmann, momentum and Maxwell is obtained, respectively as follows
(23)$$\frac{d^{2}{\bar{\psi }}_{n}\left(\bar{y}\right)}{d{\bar{y}}^{2}}={\bar{\kappa }}_{n}^{2}{\bar{\psi }}_{n}\left(\bar{y}\right),$$
(24)$${\bar{\rho }}_{n}{\bar{\lambda }}_{n}\frac{\partial^{2}{\bar{u}}_{n}\left(\bar{y},\bar{t}\right)}{\partial {\bar{t}}^{2}}+{\bar{\rho }}_{n}\frac{\partial {\bar{u}}_{n}\left(\bar{y},\bar{t}\right)}{\partial \bar{t}}={\bar{\eta }}_{n}\frac{\partial^{2}{\bar{u}}_{n}\left(\bar{y},\bar{t}\right)}{\partial {\bar{y}}^{2}}+{\bar{\epsilon }}_{n}{\bar{\kappa }}_{n}^{2}{\bar{\psi }}_{n}\left(\bar{y}\right)$$
and
(25)$${\bar{\tau }}_{xy,n}\left(\bar{y},\bar{t}\right)+{\bar{\lambda }}_{n}\frac{\partial {\bar{\tau }}_{xy,n}\left(\bar{y},\bar{t}\right)}{\partial \bar{t}}=-{\bar{\eta }}_{n}\frac{\partial {\bar{u}}_{n}\left(\bar{y},\bar{t}\right)}{\partial \bar{y}}.$$

The Equation (Equation 22) is also replaced in all boundary conditions for $\bar{t}>0$. From Equation (Equation 14) for the bottom wall of microchannel at $\bar{y}=0$, yields
(26)$${\bar{\psi }}_{1}\left(\bar{y}=0\right)=1,\;{\bar{u}}_{1}\left(\bar{y}=0,\bar{t}\right)=0,$$
respectively, from Equations (Equation 15)–(Equation 19) for each liquid-liquid interface at $\bar{y}={\bar{y}}_{n=1,2,3,...,i-1}$, leaves the following set of dimensionless equations
(27)$${\bar{\psi }}_{n+1}\left(\bar{y}\right)-{\bar{\psi }}_{n}\left(\bar{y}\right)=\Delta {\bar{\psi }}_{n},$$
(28)$${\bar{\epsilon }}_{n+1}\frac{d{\bar{\psi }}_{n+1}\left(\bar{y}\right)}{d\bar{y}}={\bar{\epsilon }}_{n}\frac{d{\bar{\psi }}_{n}\left(\bar{y}\right)}{d\bar{y}},$$
(29)$${\bar{u}}_{n+1}\left(\bar{y},\bar{t}\right)={\bar{u}}_{n}\left(\bar{y},\bar{t}\right)$$
and
(30)$${\bar{\tau }}_{xy,n+1}\left(\bar{y},\bar{t}\right)+{\bar{\epsilon }}_{n+1}\frac{d{\bar{\psi }}_{n+1}\left(\bar{y}\right)}{d\bar{y}}={\bar{\tau }}_{xy,n}\left(\bar{y},\bar{t}\right)+{\bar{\epsilon }}_{n}\frac{d{\bar{\psi }}_{n}\left(\bar{y}\right)}{d\bar{y}}=,$$
also, from Equation (Equation 20) for the top wall of microchannel at $\bar{y}=2$, yields
(31)$${\bar{\psi }}_{i}\left(\bar{y}=2\right)=1,\;{\bar{u}}_{i}\left(\bar{y}=2,\bar{t}\right)=0.$$

In the case of the initial conditions given in Equation (Equation 21), and by using Equation (Equation 22), these can be rewritten as
(32)$${\bar{u}}_{n}\left(\bar{y},\bar{t}=0\right)=0,\;{\bar{\tau }}_{xy,n}\left(\bar{y},\bar{t}=0\right)=0,\;{\left.\frac{\partial {\bar{u}}_{n}}{\partial \bar{t}}\right|}_{\bar{y},\bar{t}=0}=0,$$
for $0\leq \bar{y}\leq 2$.

The dimensionless parameters in this section are defined as
(33)$$\begin{array}{c} {\bar{\kappa }}_{n}=\frac{H}{\kappa^{-1}},\;{\bar{\rho }}_{n}=\frac{\rho_{n}}{\rho_{\mathrm{ref}}},\;{\bar{\epsilon }}_{n}=\frac{\epsilon_{n}}{\epsilon_{\mathrm{ref}}},\;{\bar{\eta }}_{n}=\frac{\eta_{0,n}}{\eta_{\mathrm{ref}}},\\ {\bar{\lambda }}_{n}=\frac{\eta_{\mathrm{ref}}\lambda_{n}}{\rho_{\mathrm{ref}}H^{2}},\;{\bar{y}}_{n}=\frac{y_{n}}{H},\;\Delta {\bar{\psi }}_{n}=\frac{\Delta \psi_{n}}{\zeta_{w}}, \end{array}$$
where ${\bar{\kappa }}_{n}$ are the ratios between the microchannel height to the Debye lengths or also known as electrokinetic parameters, ${\bar{\rho }}_{n}$ are the densities ratios, ${\bar{\epsilon }}_{n}$ are the dielectric permittivities ratios, ${\bar{\eta }}_{n}$ are the viscosity ratios and ${\bar{\lambda }}_{n}$ are the dimensionless relaxation times. On the other hand, ${\bar{y}}_{n}$ are the interface positions and $\Delta {\bar{\psi }}_{n}$ are the potential differences; these two dimensionless parameters ranging from n=1 to n=i−1.

## 3. Solution Methodology

### 3.1. Electric Potential Distribution

The Poisson-Boltzmann equation for the electric potential, Equation (Equation 23), has a well-known solution that, in terms of *n*-layers of fluid is given by
(34)$${\bar{\psi }}_{n}\left(\bar{y}\right)=C_{2n-1}e^{{\bar{\kappa }}_{n}\bar{y}}+C_{2n}e^{-{\bar{\kappa }}_{n}\bar{y}},$$
where $C_{2n-1}$ and $C_{2n}$ are integration constants that are determined by applying the corresponding boundary conditions at solid-liquid and liquid-liquid interfaces given in Equations (Equation 26)–(Equation 28) and (Equation 31)–(Equation 34). As result, the following equation system is obtained
(35)$$\begin{array}{cc} C_{1}+C_{2} & =1,\\ C_{3}e^{{\bar{\kappa }}_{2}{\bar{y}}_{1}}+C_{4}e^{-{\bar{\kappa }}_{2}{\bar{y}}_{1}}-C_{1}e^{{\bar{\kappa }}_{1}{\bar{y}}_{1}}-C_{2}e^{-{\bar{\kappa }}_{1}{\bar{y}}_{1}} & =\Delta {\bar{\psi }}_{1},\\ {\bar{\epsilon }}_{2}\left[C_{3}{\bar{\kappa }}_{2}e^{{\bar{\kappa }}_{2}{\bar{y}}_{1}}-C_{4}{\bar{\kappa }}_{2}e^{-{\bar{\kappa }}_{2}{\bar{y}}_{1}}\right]-{\bar{\epsilon }}_{1}\left[C_{1}{\bar{\kappa }}_{1}e^{{\bar{\kappa }}_{1}{\bar{y}}_{1}}-C_{2}{\bar{\kappa }}_{1}e^{-{\bar{\kappa }}_{1}{\bar{y}}_{1}}\right] & =0,\\ C_{5}e^{{\bar{\kappa }}_{3}{\bar{y}}_{2}}+C_{6}e^{-{\bar{\kappa }}_{3}{\bar{y}}_{2}}-C_{3}e^{{\bar{\kappa }}_{2}{\bar{y}}_{2}}-C_{4}e^{-{\bar{\kappa }}_{2}{\bar{y}}_{2}} & =\Delta {\bar{\psi }}_{2},\\ {\bar{\epsilon }}_{3}\left[C_{5}{\bar{\kappa }}_{3}e^{{\bar{\kappa }}_{3}{\bar{y}}_{2}}-C_{6}{\bar{\kappa }}_{3}e^{-{\bar{\kappa }}_{3}{\bar{y}}_{2}}\right]-{\bar{\epsilon }}_{2}\left[C_{3}{\bar{\kappa }}_{2}e^{{\bar{\kappa }}_{2}{\bar{y}}_{2}}-C_{4}{\bar{\kappa }}_{2}e^{-{\bar{\kappa }}_{2}{\bar{y}}_{2}}\right] & =0,\\ \vdots & \\ C_{2i-1}e^{{\bar{\kappa }}_{i}{\bar{y}}_{i-1}}+C_{2i}e^{-{\bar{\kappa }}_{i}{\bar{y}}_{i-1}}-C_{2(i-1)-1}e^{{\bar{\kappa }}_{i-1}{\bar{y}}_{i-1}}-C_{2(i-1)}e^{-{\bar{\kappa }}_{i-1}{\bar{y}}_{i-1}} & =\Delta {\bar{\psi }}_{i-1},\\ {\bar{\epsilon }}_{i}\left[C_{2(i)-1}{\bar{\kappa }}_{i}e^{{\bar{\kappa }}_{i}{\bar{y}}_{i-1}}-C_{2(i)}{\bar{\kappa }}_{i}e^{-{\bar{\kappa }}_{i}{\bar{y}}_{i-1}}\right]-{\bar{\epsilon }}_{i-1}\left[C_{2(i-1)-1}{\bar{\kappa }}_{i-1}e^{{\bar{\kappa }}_{i-1}{\bar{y}}_{i-1}}-C_{2(i-1)}{\bar{\kappa }}_{i-1}e^{-{\bar{\kappa }}_{i-1}{\bar{y}}_{i-1}}\right] & =0,\\ C_{2i-1}e^{2{\bar{\kappa }}_{i}}+C_{2i}e^{-2{\bar{\kappa }}_{i}} & =1. \end{array}$$

The above system of linear algebraic equations contains the same number of variables as the equations. Hence, the integration constants $C_{2n-1}$ and $C_{2n}$ are solved by the matrix inverse method [65].

### 3.2. Velocity Distribution

To obtain the dimensionless velocity distribution, the Laplace transforms are defined as
(36)$$U_{n}\left(\bar{y},s\right)=L\left\{{\bar{u}}_{n}\left(\bar{y},\bar{t}\right)\right\}=\int_{0}^{\infty }{\bar{u}}_{n}\left(\bar{y},\bar{t}\right)e^{-s\bar{t}}d\bar{t}$$
and for the shear stress the following relationship is also used
(37)$${\tilde{\tau }}_{xy,n}\left(\bar{y},s\right)=L\left\{{\bar{\tau }}_{xy,n}\left(\bar{y},\bar{t}\right)\right\}=\int_{0}^{\infty }{\bar{\tau }}_{xy,n}\left(\bar{y},\bar{t}\right)e^{-s\bar{t}}d\bar{t}.$$

Equations (Equation 36) and (Equation 37) were applied to the momentum and constitutive equations, Equation (Equation 24) and (Equation 25), yielding
(38)$$\begin{array}{cc} {\bar{\lambda }}_{n}{\bar{\rho }}_{n} & \left[s^{2}U_{n}\left(\bar{y},s\right)-s{\bar{u}}_{n}\left(\bar{y},\bar{t}=0\right)-{\left.\frac{\partial {\bar{u}}_{n}}{\partial \bar{t}}\right|}_{\bar{y},\bar{t}=0}\right]\\ \; & +{\bar{\rho }}_{n}\left[s^{2}U_{n}\left(\bar{y},s\right)-s{\bar{u}}_{n}\left(\bar{y},\bar{t}=0\right)\right]={\bar{\eta }}_{n}\frac{\partial^{2}U_{n}\left(\bar{y},s\right)}{\partial {\bar{y}}^{2}}+\frac{{\bar{\epsilon }}_{n}{\bar{\kappa }}_{n}^{2}}{{\bar{\eta }}_{n}s}{\bar{\psi }}_{n}\left(\bar{y}\right) \end{array}$$
and respectively
(39)$${\tilde{\tau }}_{xy,n}\left(\bar{y},s\right)+{\bar{\lambda }}_{n}\left[s{\tilde{\tau }}_{xy,n}\left(\bar{y},s\right)-{\bar{\tau }}_{xy,n}\left(\bar{y},\bar{t}=0\right)\right]=-{\bar{\eta }}_{n}\frac{\partial U_{n}}{\partial \bar{y}}.$$

Satisfying the initial conditions given in Equations (Equation 32), (Equation 38) and (Equation 39) can be rewritten as
(40)$$\frac{\partial^{2}U_{n}\left(\bar{y},s\right)}{\partial {\bar{y}}^{2}}-\alpha_{n}^{2}U_{n}\left(\bar{y},s\right)=\beta_{n}{\bar{\psi }}_{n}\left(\bar{y}\right)$$
and
(41)$${\tilde{\tau }}_{xy,n}\left(\bar{y},s\right)=-\gamma_{n}\frac{\partial U_{n}\left(\bar{y},s\right)}{\partial \bar{y}},$$
where $\alpha_{n}^{2}=\left({\bar{\rho }}_{n}s/{\bar{\eta }}_{n}\right)\left({\bar{\lambda }}_{n}s+1\right)$, $\beta_{n}=-{\bar{\epsilon }}_{n}{\bar{\kappa }}_{n}^{2}/{\bar{\eta }}_{n}s$, and $\gamma_{n}={\bar{\eta }}_{n}/\left(1+{\bar{\lambda }}_{n}s\right)$. To obtain the corresponding boundary conditions to solve the momentum Equation (Equation 40), the Laplace transforms in Equations (Equation 36) and (Equation 37) are applied in Equations (Equation 26), (Equation 29)–(Equation 31), yielding for the bottom wall of microchannel at $\bar{y}=0$
(42)$$U_{1}\left(\bar{y}=0,s\right)=0,$$
in each liquid-liquid interface at $\bar{y}={\bar{y}}_{n=1,2,3,...,i-1}$ and in addition with aid of Equation (Equation 41), yields
(43)$$U_{n}\left(\bar{y}={\bar{y}}_{n},s\right)=U_{n+1}\left(\bar{y}={\bar{y}}_{n},s\right),$$
(44)$$-\gamma_{n}\frac{\partial U_{n}}{\partial \bar{y}}+\frac{{\bar{\epsilon }}_{n}}{s}\frac{d{\bar{\psi }}_{n}}{d\bar{y}}=-\gamma_{n+1}\frac{\partial U_{n+1}}{\partial \bar{y}}+\frac{{\bar{\epsilon }}_{n+1}}{s}\frac{d{\bar{\psi }}_{n+1}}{d\bar{y}},$$
and finally, for the boundary condition for the top wall of microchannel at $\bar{y}=2$ is obtained that
(45)$$U_{i}\left(\bar{y}=2,s\right)=0.$$

Therefore, the mathematical model for electroosmotic flow of multi-layer immiscible Maxwell fluids in the space of the Laplace transform is composed by Equations (Equation 40), (Equation 42)–(Equation 45). Being Equation (Equation 40) an nonhomogeneous ordinary differential equation, its solution is the superposition of a homogeneous solution $U_{h,n}\left(\bar{y},s\right)$ and a particular solution $U_{p,n}\left(\bar{y},s\right)$ as follows
(46)$$U_{n}\left(\bar{y},s\right)=U_{h,n}\left(\bar{y},s\right)+U_{p,n}\left(\bar{y},s\right).$$

The homogeneous solution and the particular one, are the following equations, respectively
(47)$$U_{h,n}\left(\bar{y},s\right)=A_{n}e^{\alpha_{n}\bar{y}}+B_{n}e^{-\alpha_{n}\bar{y}}$$
and
(48)$$U_{p,n}\left(\bar{y},s\right)=D_{n}e^{{\bar{\kappa }}_{n}\bar{y}}+E_{n}e^{-{\bar{\kappa }}_{n}\bar{y}}$$
where $A_{n}$, $B_{n}$, $D_{n}$, and $E_{n}$ are constant to be determined.

The constants $D_{n}$ and $E_{n}$ are obtained by the substitution of Equation (Equation 48) into the Equation (Equation 40), yielding
(49)$$D_{n}=\frac{\beta_{n}C_{n}}{{\bar{\kappa }}_{n}^{2}-\alpha_{n}^{2}},\;E_{n}=\frac{\beta_{n}C_{n+1}}{{\bar{\kappa }}_{n}^{2}-\alpha_{n}^{2}}.$$

Considering the constant values of the previous equation and Equation (Equation 46), the dimensionless velocity distribution of each fluid layer of electroosmotic flow is
(50)$$U_{n}\left(\bar{y},s\right)=A_{n}e^{\alpha_{n}\bar{y}}+B_{n}e^{-\alpha_{n}\bar{y}}+D_{n}e^{{\bar{\kappa }}_{n}\bar{y}}+E_{n}e^{-{\bar{\kappa }}_{n}\bar{y}}.$$

To find the constants $A_{n}$ and $B_{n}$, it is necessary to apply the boundary conditions given from Equations (Equation 42)–(Equation 45) to Equation (Equation 50), and with aid of Equation (Equation 34), the following system of linear algebraic equations is obtained
(51)$$\begin{array}{cc} \; & A_{1}+B_{1}+D_{1}+E_{1}=0,\\ \; & A_{1}e^{\alpha_{1}{\bar{y}}_{1}}+B_{1}e^{-\alpha_{1}{\bar{y}}_{1}}+D_{1}e^{{\bar{\kappa }}_{1}{\bar{y}}_{1}}+E_{1}e^{{\bar{\kappa }}_{1}{\bar{y}}_{1}}=A_{2}e^{\alpha_{2}{\bar{y}}_{1}}+B_{2}e^{-\alpha_{2}{\bar{y}}_{1}}+D_{2}e^{{\bar{\kappa }}_{2}{\bar{y}}_{1}}+E_{2}e^{{\bar{\kappa }}_{2}{\bar{y}}_{1}},\\ \; & \gamma_{1}\left[A_{1}\alpha_{1}e^{\alpha_{1}{\bar{y}}_{1}}-B_{1}\alpha_{1}e^{-\alpha_{1}{\bar{y}}_{1}}+\right.\left.D_{1}{\bar{\kappa }}_{n}e^{{\bar{\kappa }}_{1}{\bar{y}}_{1}}-E_{1}{\bar{\kappa }}_{1}e^{-{\bar{\kappa }}_{1}{\bar{y}}_{1}}\right]+\frac{{\bar{\epsilon }}_{1}}{s}\left[C_{1}{\bar{\kappa }}_{1}e^{{\bar{\kappa }}_{1}{\bar{y}}_{1}}-C_{2}{\bar{\kappa }}_{1}e^{-{\bar{\kappa }}_{1}{\bar{y}}_{1}}\right]=\\ \; & \gamma_{2}\left[A_{2}\alpha_{2}e^{\alpha_{2}{\bar{y}}_{1}}-B_{2}\alpha_{2}e^{-\alpha_{2}{\bar{y}}_{1}}+\right.\left.D_{2}{\bar{\kappa }}_{2}e^{{\bar{\kappa }}_{2}{\bar{y}}_{1}}-E_{2}{\bar{\kappa }}_{2}e^{-{\bar{\kappa }}_{2}{\bar{y}}_{1}}\right]+\frac{{\bar{\epsilon }}_{2}}{s}\left[C_{3}{\bar{\kappa }}_{2}e^{{\bar{\kappa }}_{2}{\bar{y}}_{1}}-C_{4}{\bar{\kappa }}_{2}e^{-{\bar{\kappa }}_{2}{\bar{y}}_{1}}\right],\\ \; & A_{2}e^{\alpha_{2}{\bar{y}}_{2}}+B_{2}e^{-\alpha_{2}{\bar{y}}_{2}}+D_{2}e^{{\bar{\kappa }}_{2}{\bar{y}}_{2}}+E_{2}e^{{\bar{\kappa }}_{2}{\bar{y}}_{2}}=A_{3}e^{\alpha_{3}{\bar{y}}_{2}}+B_{3}e^{-\alpha_{3}{\bar{y}}_{2}}+D_{3}e^{{\bar{\kappa }}_{3}{\bar{y}}_{2}}+E_{3}e^{{\bar{\kappa }}_{3}{\bar{y}}_{2}},\\ \; & \gamma_{2}\left[A_{2}\alpha_{2}e^{\alpha_{2}{\bar{y}}_{2}}-B_{2}\alpha_{2}e^{-\alpha_{2}{\bar{y}}_{2}}+\right.\left.D_{2}{\bar{\kappa }}_{2}e^{{\bar{\kappa }}_{2}{\bar{y}}_{2}}-E_{2}{\bar{\kappa }}_{2}e^{-{\bar{\kappa }}_{2}{\bar{y}}_{2}}\right]+\frac{{\bar{\epsilon }}_{2}}{s}\left[C_{3}{\bar{\kappa }}_{2}e^{{\bar{\kappa }}_{2}{\bar{y}}_{2}}-C_{4}{\bar{\kappa }}_{2}e^{-{\bar{\kappa }}_{2}{\bar{y}}_{2}}\right]=\\ \; & \gamma_{3}\left[A_{3}\alpha_{3}e^{\alpha_{3}{\bar{y}}_{2}}-B_{3}\alpha_{3}e^{-\alpha_{3}{\bar{y}}_{2}}+\right.\left.D_{3}{\bar{\kappa }}_{3}e^{{\bar{\kappa }}_{3}{\bar{y}}_{2}}-E_{3}{\bar{\kappa }}_{3}e^{-{\bar{\kappa }}_{3}{\bar{y}}_{2}}\right]+\frac{{\bar{\epsilon }}_{3}}{s}\left[C_{5}{\bar{\kappa }}_{3}e^{{\bar{\kappa }}_{3}{\bar{y}}_{2}}-C_{6}{\bar{\kappa }}_{3}e^{-{\bar{\kappa }}_{3}{\bar{y}}_{2}}\right],\\ \; & \;\vdots \\ \; & A_{i-1}e^{\alpha_{i-1}{\bar{y}}_{i-1}}+B_{i-1}e^{-\alpha_{i-1}{\bar{y}}_{i-1}}+D_{i-1}e^{{\bar{\kappa }}_{i-1}{\bar{y}}_{i-1}}+E_{i-1}e^{{\bar{\kappa }}_{i-1}{\bar{y}}_{i-1}}=\\ \; & A_{i}e^{\alpha_{i}{\bar{y}}_{i-1}}+B_{i}e^{-\alpha_{i}{\bar{y}}_{i-1}}+D_{i}e^{{\bar{\kappa }}_{i}{\bar{y}}_{i-1}}+E_{i}e^{{\bar{\kappa }}_{i}{\bar{y}}_{i-1}},\\ \; & \gamma_{i-1}\left[A_{i-1}\alpha_{i-1}e^{\alpha_{i-1}{\bar{y}}_{i-1}}-B_{i-1}\alpha_{i-1}e^{-\alpha_{i-1}{\bar{y}}_{i-1}}+\right.\left.D_{i-1}{\bar{\kappa }}_{i-1}e^{{\bar{\kappa }}_{i-1}{\bar{y}}_{i-1}}-E_{i-1}{\bar{\kappa }}_{i-1}e^{-{\bar{\kappa }}_{i-1}{\bar{y}}_{i-1}}\right]\\ \; & \;+\frac{{\bar{\epsilon }}_{i-1}}{s}\left[C_{2(i-1)-1}{\bar{\kappa }}_{i-1}e^{{\bar{\kappa }}_{i-1}{\bar{y}}_{i-1}}-\right.\left.C_{2(i-1)}{\bar{\kappa }}_{i-1}e^{-{\bar{\kappa }}_{i-1}{\bar{y}}_{i-1}}\right]=\gamma_{i}\left[A_{i}\alpha_{i}e^{\alpha_{i}{\bar{y}}_{i-1}}-B_{i}\alpha_{i}e^{-\alpha_{i}{\bar{y}}_{i-1}}\right.\\ \; & \;+\left.D_{i}{\bar{\kappa }}_{i}e^{{\bar{\kappa }}_{i}{\bar{y}}_{i-1}}-E_{i}{\bar{\kappa }}_{i}e^{-{\bar{\kappa }}_{i}{\bar{y}}_{i-1}}\right]+\frac{{\bar{\epsilon }}_{i}}{s}\left[C_{2(i)-1}{\bar{\kappa }}_{i}e^{{\bar{\kappa }}_{i}{\bar{y}}_{i-1}}-C_{2(i)}{\bar{\kappa }}_{i}e^{-{\bar{\kappa }}_{i}{\bar{y}}_{i-1}}\right],\\ \; & A_{i}e^{2\alpha_{i}}+B_{i}e^{-2\alpha_{i}}+D_{i}e^{2{\bar{\kappa }}_{i}}+E_{i}e^{-2{\bar{\kappa }}_{i}}=0, \end{array}$$
which has been solved using the inverse matrix method in a process analogous to that of the electric potential distribution. Therefore, the constants $D_{n}$ and $E_{n}$ in Equation (Equation 49), and the constants $A_{n}$ and $B_{n}$ found through Equation (Equation 51), are replaced into Equation (Equation 50), where the inverse Laplace transform is numerically applied to solve the velocity distribution in this electroosmotic flow. To this, the method based on concentrated matrix exponential (CME) distributions is used [66]; in this framework, a finite linear combination of the transform values approximates $\bar{u}$, via
(52)$${\bar{u}}_{n}\left(\bar{y},\bar{t}\right)\approx {\bar{u}}_{n}\left(\bar{y},\bar{t},M\right)=\frac{1}{\bar{t}}\sum_{k=1}^{M}\omega_{k}U_{n}\left(\bar{y},\frac{\theta_{k}}{\bar{t}}\right),$$
where $\omega_{1}$ and $\theta_{1}$ are real coefficients, and from $\omega_{2}$ to $\omega_{M}$, and from $\theta_{2}$ to $\theta_{M}$ are (M−1)/2 complex conjugate pairs that the authors in Horváth et al. [66] provide. Here, M=50.

## 4. Results and Discussion

The dimensionless parameters used in the results are estimated with an appropriate combination of the following geometric dimensions, physical properties, and electrostatic interfacial conditions in the range of: 0.1≤2H≤10μm, $1\leq \kappa_{n}^{-1}\leq 200$ nm, $\rho_{n}\approx 1000$ kg m${}^{-3}$, $10^{-4}\;\leq \;\eta_{0,n}\;\leq \;10^{-2}$ kg m${}^{-1}$s${}^{-1}$, $E_{x}\leq 10^{4}$ Vm${}^{-1}$, $10^{-11}\leq \epsilon_{n}\leq 10^{-9}$ C V${}^{-1}$ m${}^{-1}$, $z_{n}∼O\left(1\right)$, $0\leq \lambda_{n}\leq 0.01$ s, $-50\;\leq \;\zeta_{w}\;\leq \;50$ mV, $-12.5\leq \Delta \psi_{n}\leq 12.5$ mV; also, the values of the constants $k_{B}=1.381\times 10^{23}$ J K${}^{-1}$ and $e=1.602\times 10^{-19}$ C, are considered.

### 4.1. Solution Validation

To validate the performance of the semi-analytical solution found in this work for the transient velocity distribution, a comparison was made with two investigations reported by the scientific community, considering the transport of Newtonian and Maxwell fluids, respectively. In the first case, in the research carried out by Yang et al. [10], they model an electroosmotic flow of an aqueous 1:1 electrolyte (NaCl) in a slit microchannel with the following physical properties: a density of ρ = 998 kg m${}^{-3}$, a viscosity of η=0.90×10${}^{-3}$ kg m ${}^{-1}$ s${}^{-1}$, a relative electrical permittivity of $\epsilon_{r}$ = 80, and a concentration of 10${}^{-4}$ M, at a temperature of *T* = 298 K; additionally, the microchannel size and the wall zeta potential were set at 2H = 10 μm and 50 mV, respectively. With that set of values, the following electrokinetic parameter is obtained, ${\bar{\kappa }}_{n}$=164.5, and the dimensionless times to evaluate the velocity profiles are $\bar{t}$ = 0.0036, 0.036, 0.36 and 3.6 (=0.1, 1, 10 and 100 μs). Therefore, by comparing the work of Yang et al. [10] with the present investigation for three immiscible fluid layers, in Figure 2, an excellent convergence between their results is shown.

In the second case, the analytical solution for the dimensionless velocity profiles obtained by Escandón et al. [16] on the transient electroosmotic flow with Maxwell fluids, are compared with the present study, as shown in Figure 3. Here, the electrokinetic parameter takes a value of ${\bar{\kappa }}_{n}=20$ and the viscoelastic behavior of fluids is presented trough the two dimensionless relaxations time values of ${\bar{\lambda }}_{n}=0.12$ and ${\bar{\lambda }}_{n}=2.5$ in Figure 3a,b, respectively, finding a very good match between the results.

### 4.2. Velocity Profiles

Figure 4 shows the dimensionless electric potential profile, ${\bar{\psi }}_{n}$, and the start-up of the electroosmotic flow velocity, ${\bar{u}}_{n}$, of three layers (n= 3) of immiscible Maxwell fluids in a slit microchannel, both variables as a function of the transverse coordinate $\bar{y}$ and for three different values of the potential difference $\Delta {\bar{\psi }}_{n}\left(=0.5,0,-0.5\right)$. The interfaces between fluids have been placed in ${\bar{y}}_{1}=2/3$ and ${\bar{y}}_{2}=4/3$, respectively, and the other dimensionless parameters are specified in the caption of the figure. Because of the high ionic concentration in the diffuse layers within the electric double layers formed in the solid-liquid interfaces in the system, there is a higher magnitude of the electric potential in these zones. On the other hand, in Figure 4a with $\Delta {\bar{\psi }}_{n}=0.5$ and Figure 4c with $\Delta {\bar{\psi }}_{n}=-0.5$, can be appreciated an electric slip at liquid-liquid interfaces due to counterions concentrations in each side of interfaces, while for Figure 4b with $\Delta {\bar{\psi }}_{n}=0$, the classical null distribution of electrical potential is recovered outside of the electrical double layers on the walls. Here, the potential difference or electric slip between immiscible layers is proportional to the difference in the magnitude of $\Delta {\bar{\psi }}_{n}$ given by Equation (Equation 27) at each interface, and the sign gives the orientation of the counterions and electric potential distribution. Regarding the velocity, in each Figure 4a–c, are shown the evolution of the velocity profiles since an early time of $\bar{t}=0.05$ to the steady-state when $\bar{t}\to \infty$. As can be seen, for the early times, the fluids movement beginning from the Debye length in the solid-liquid and liquid-liquid interfaces due to electroosmotic effects, transmitting the movement by viscous forces to the rest of the fluid layers as time progresses. The influence of the potential difference on velocity development is clear when comparing Figure 4a,c with Figure 4b, producing great disturbances and steep velocity gradients in the flow velocity.

Figure 5 shows the elastic behavior of the Maxwell fluids via the dimensionless relaxation time on the flow dynamics. In this figure are presented three cases for the velocity profiles evolution, in Figure 5a–c, the selected dimensionless relaxation times values are ${\bar{\lambda }}_{n}=0.1$, ${\bar{\lambda }}_{n}=2$ and ${\bar{\lambda }}_{n}\;=\;10$, respectively. In these figures, it is noticeable that the start-up of fluids is more slowly as the dimensionless relaxation time increases, due to the memory effects of the viscoelastic fluids also increase, delaying the start of movement of fluids. Hence, in the case of Figure 5a the time to reach the steady-state when $\bar{t}\to \infty$ is much shorter than Figure 5c; in this context, the velocity magnitude and oscillatory behavior are greater for the case with ${\bar{\lambda }}_{n}=10$ than for ${\bar{\lambda }}_{n}=0.1$, because there is a more severe competition between the viscous and the elastic forces in the first case, that is, with ${\bar{\lambda }}_{n}=10$. For all cases in Figure 5a–c, reverse flow is produced in early times due to the potential difference at liquid-liquid interfaces.

In Figure 6 is represented the evolution of the velocity profiles of three fluid layers as a function of the transverse coordinate, and three combinations of the viscosities ratios ${\bar{\eta }}_{n}\left(=0.7,2,6\right)$. It is evident that in Figure 6a, being the case with the lower viscosity fluid layers with ${\bar{\eta }}_{n}=0.7$, the flow has the highest magnitude of velocity profiles; this can be corroborated if it is compared with the case of Figure 6c with a viscosity of ${\bar{\eta }}_{n}=6$ where the fluids have a greater resistance to flow and a lower magnitude of velocity. In all Figure 6a–c, the oscillatory behavior from elastic effects of Maxwell fluids is maintained.

Figure 7 shows a wide combination of all dimensionless parameters studied in the present work. Here is observed the response of the dimensionless electric potential and velocity profiles for the electroosmotic flow of four layers (n=4) of immiscible Maxwell fluids, under different values of electrokinetic parameters (${\bar{\kappa }}_{n=1,2,3,4}=10,20,30,40$) and relaxation times (${\bar{\lambda }}_{n=1,2,3,4}=0.1,2,1,10$). The velocity evolution goes from the time $\bar{t}=0.1$ to the steady-state when $\bar{t}\to \infty$. It can be seen a constant and gradual velocity evolution for the fluid 1, due to the small value of the relaxation time while for a higher relaxation time value the velocity profile oscillates continuously until reaching the steady-state resulting in stronger memory effects from the viscoelastic fluid such is the case of the fluid 4. Regarding the electrokinetic parameter effect, a value of ${\bar{\kappa }}_{1}=10$ in fluid 1 yields a parabolic shape of the velocity profiles due to the low ionic concentration of the buffer solution, producing a thick electric double layer; contrary the aforementioned, as the electrokinetic parameter grows in fluid 4 to take the value of ${\bar{\kappa }}_{4}=40$, the electric double layer becomes thinner and results in more slanted and straighter velocity profiles.

## 5. Tracking of the Velocity

In Figure 8 the tracking of the dimensionless velocity on the transverse coordinate $\bar{y}=1$ as a function of the dimensionless time is presented. In all cases of the sub-figures contained in Figure 8, the start-up of movement of fluid(s) in the center of the microchannel is delayed by memory effects due to its viscoelastic properties. In general, after overcoming these memory effects, a sudden and severe increase in velocity occurs, beginning a continuous oscillatory movement of increasing and decreasing velocity until steady-state is reached. The results presented in Figure 8a–c are taken from Figure 4, Figure 5 and Figure 6, respectively. It is clear from Figure 8a,d,f, that the time it will take for the fluids to reach the steady-state is independent of the dimensionless parameters $\Delta {\bar{\psi }}_{n}$, ${\bar{\rho }}_{n}$ and ${\bar{\kappa }}_{n}$, respectively. However, from Figure 8b, the time to reach the steady-state in the multi-layer electroosmotic flow is strongly dependent of the relaxation time, where for ${\bar{\lambda }}_{n}=0.1$, ${\bar{\lambda }}_{n}=2$ and ${\bar{\lambda }}_{n}=10$ the dimensionless times to reach the steady-state are ${\bar{t}}_{ss}\approx 2.27$, ${\bar{t}}_{ss}\approx 36.36$ and ${\bar{t}}_{ss}\approx 181.82$, respectively, these time results are due to the increases of fluid elasticity via the parameter ${\bar{\lambda }}_{n}$. In this context, from Figure 8c the time to reach the steady-state in the flow is also dependent of the viscosity ratios ${\bar{\eta }}_{n}$, being the less viscous fluids those that take longer to reach that regime with ${\bar{\eta }}_{n}=0.7$ in a time of ${\bar{t}}_{ss}\approx 20$, while on the contrary case, with ${\bar{\eta }}_{n}=6$, the steady-state is reached in a shorter time with ${\bar{t}}_{ss}\approx 12.73$, due to the increase of the viscous forces and the corresponding faster braking of the flow. Furthermore, it can be seen in Figure 8e that multi-layer flows with three or more fluids take less time to reach the permanent regime due to the combined effects at the liquid-liquid interfaces. Finally, the time to reach the steady-state regime is established for the present work as the time in which the absolute value of the velocity difference between two immediate times at the same position is less than $10^{-3}$.

## 6. Conclusions

In this investigation, a semi-analytical solution of the start-up of the electroosmotic flow of multi-layer immiscible Maxwell fluids in a slit microchannel was obtained. In the parametric study, different fluid properties, geometric characteristics of the number and thickness of fluid layers, and electrostatic boundary conditions at liquid-liquid interfaces were considered. The electrostatic conditions from the electric double layers between the fluids via the potential differences and electro-viscous shear stresses, break the continuity of the electric potential distribution and produce significant changes in the velocity profiles in these zones. Regarding the dimensionless relaxation time effects on the velocity profiles, as this parameter increases a longer oscillatory behavior is caused by the memory effects of Maxwell fluids. Likewise, the magnitude of the flow velocity will significantly reduce with layers of more viscous fluids due to greater resistance to flow. In other results on the fluid dynamics of the multilayer electroosmotic flow, the time to reach the steady-state regime is strongly controlled by some dimensionless parameters reported here, like the relaxation times, the viscosity ratios, and the number of fluid layers.

## Figures and Tables

**Figure 1 micromachines-11-00757-f001:**
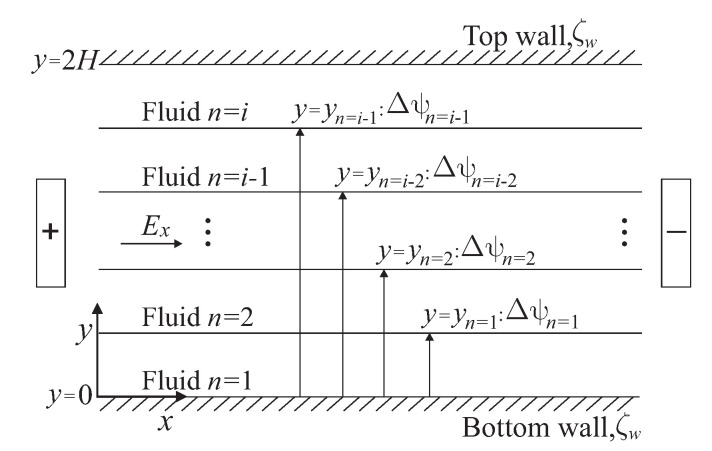
Sketch of electroosmotic flow of multi-layer immiscible fluids in a slit microchannel.

**Figure 2 micromachines-11-00757-f002:**
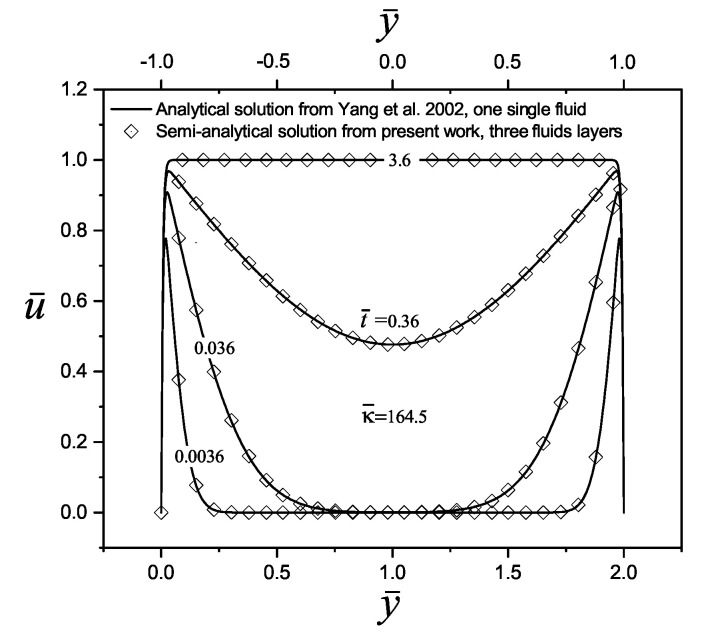
Dimensionless velocity profiles of an electroosmotic flow obtained by Yang et al. [10] with n=1, against the results of the present investigation with three fluid layers, n=3 (${\bar{y}}_{1}=$2/3 and ${\bar{y}}_{2}=$4/3). The other parameters are ${\bar{\rho }}_{n}={\bar{\eta }}_{n}={\bar{\epsilon }}_{n}=$1, and $\Delta {\bar{\psi }}_{n}=$0.

**Figure 3 micromachines-11-00757-f003:**
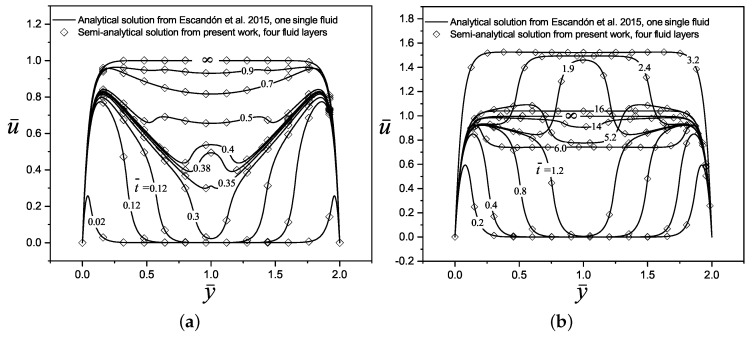
Dimensionless velocity profiles of an electroosmotic flow obtained by Escandón et al. [16] with n=1, against the results of the present investigation with four fluid layers, n=4 (${\bar{y}}_{1}=$1/2, ${\bar{y}}_{2}=$1.0, and ${\bar{y}}_{3}=$3/2). The other parameters are ${\bar{\rho }}_{n}={\bar{\eta }}_{n}={\bar{\epsilon }}_{n}=$1, and $\Delta {\bar{\psi }}_{n}=$0, for: (**a**) ${\bar{\lambda }}_{n}=0.12$ and (**b**) ${\bar{\lambda }}_{n}=2.5$.

**Figure 4 micromachines-11-00757-f004:**
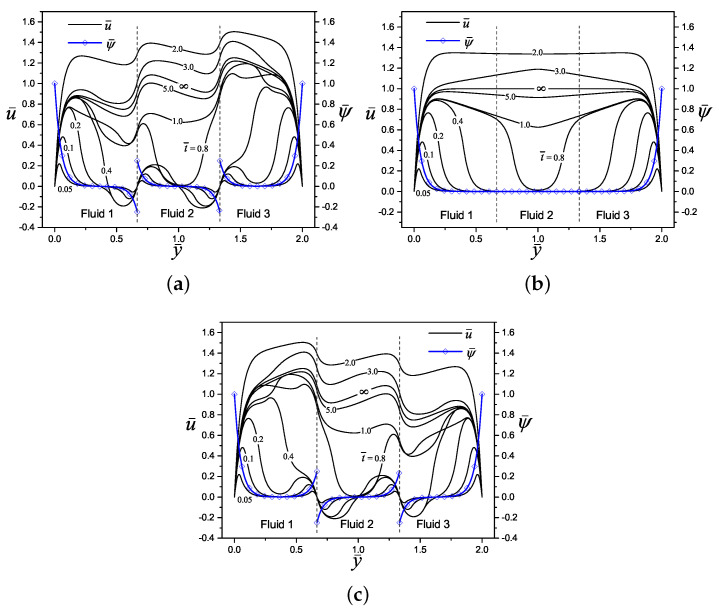
Dimensionless electric potential and velocity profiles of an electroosmotic flow with n=3, (${\bar{y}}_{1}=2/3$, ${\bar{y}}_{2}=4/3$), ${\bar{\kappa }}_{n}=20$, ${\bar{\rho }}_{n}={\bar{\eta }}_{n}={\bar{\epsilon }}_{n}={\bar{\lambda }}_{n}=$1, for: (**a**) $\Delta {\bar{\psi }}_{n}=0.5$, (**b**) $\Delta {\bar{\psi }}_{n}=0$, and (**c**) $\Delta {\bar{\psi }}_{n}=-0.5$.

**Figure 5 micromachines-11-00757-f005:**
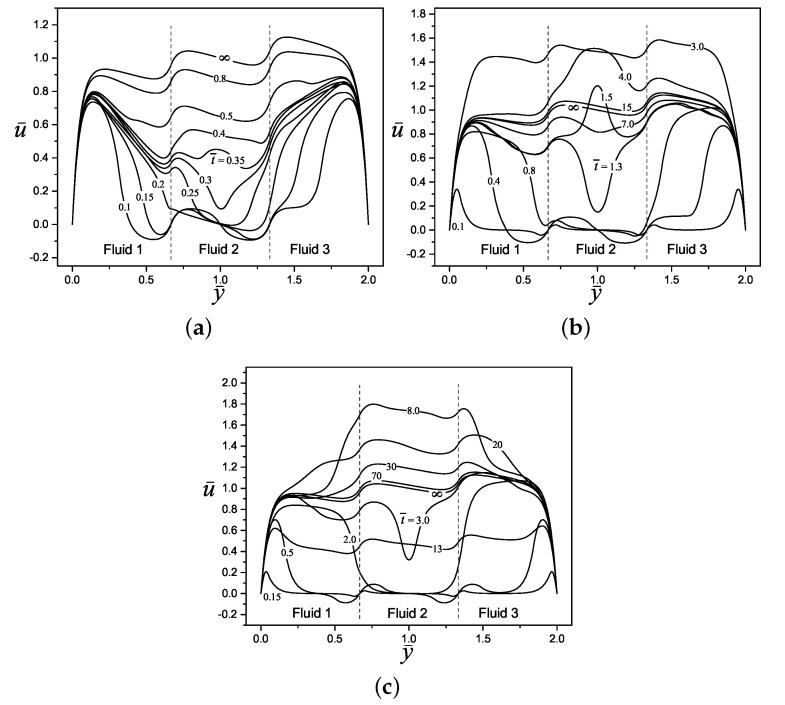
Dimensionless velocity profiles of an electroosmotic flow with n=3, (${\bar{y}}_{1}=2/3$, ${\bar{y}}_{2}=4/3$), ${\bar{\kappa }}_{n}=20$, ${\bar{\rho }}_{n}={\bar{\eta }}_{n}={\bar{\epsilon }}_{n}=$1, and $\Delta {\bar{\psi }}_{n}=0.25$ for: (**a**) ${\bar{\lambda }}_{n}=0.1$, (**b**) ${\bar{\lambda }}_{n}=2$, and (**c**) ${\bar{\lambda }}_{n}=10$.

**Figure 6 micromachines-11-00757-f006:**
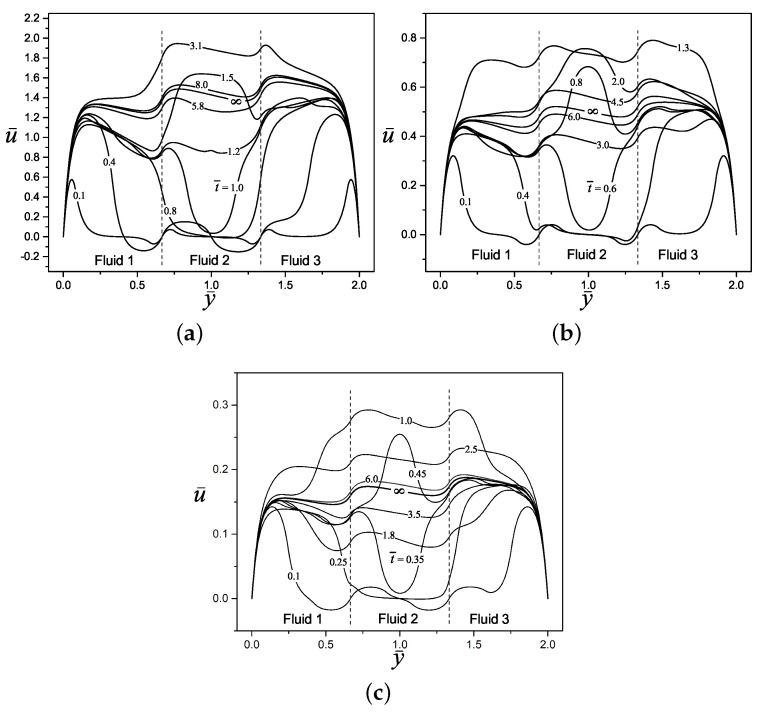
Dimensionless velocity profiles of an electroosmotic flow with n=3, (${\bar{y}}_{1}=2/3$, ${\bar{y}}_{2}=4/3$), ${\bar{\kappa }}_{n}=20$, ${\bar{\rho }}_{n}={\bar{\epsilon }}_{n}={\bar{\lambda }}_{n}=$1, and $\Delta {\bar{\psi }}_{n}=0.25$ for: (**a**) ${\bar{\eta }}_{n}=0.7$, (**b**) ${\bar{\eta }}_{n}=2$, and (**c**) ${\bar{\eta }}_{n}=6$.

**Figure 7 micromachines-11-00757-f007:**
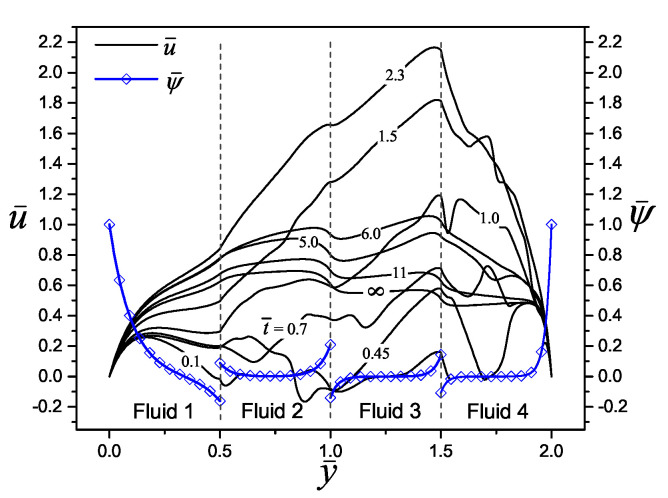
Dimensionless electric potential and velocity profiles of an electroosmotic flow with n=4, (${\bar{y}}_{1}=2/3$, ${\bar{y}}_{2}=1$, ${\bar{y}}_{3}=3/2$ ), ${\bar{\kappa }}_{1}=10$, ${\bar{\kappa }}_{2}=20$, ${\bar{\kappa }}_{3}=30$, ${\bar{\kappa }}_{4}=40$, ${\bar{\rho }}_{n}={\bar{\epsilon }}_{n}=$1, ${\bar{\lambda }}_{1}=$0.1, ${\bar{\lambda }}_{2}=$2, ${\bar{\lambda }}_{3}=$1, ${\bar{\lambda }}_{4}=$10, $\Delta {\bar{\psi }}_{1}=0.25$, $\Delta {\bar{\psi }}_{2}=-0.35$, $\Delta {\bar{\psi }}_{3}=-0.25$, and ${\bar{\eta }}_{n}=2$.

**Figure 8 micromachines-11-00757-f008:**
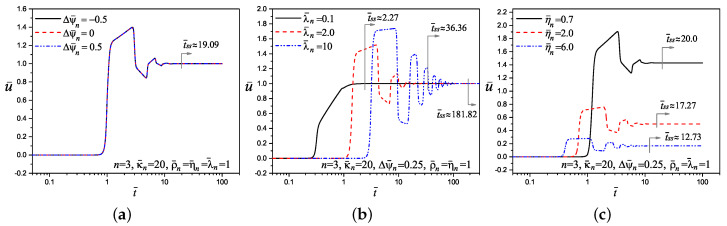
Tracking of the velocity in the multi-layer flow as a function of the dimensionless time evaluated in the center of the microchannel at $\bar{y}=1$ and with ${\bar{\epsilon }}_{n}=$1 for: (**a**) effect of $\Delta {\bar{\psi }}_{n}$ (from Figure 4), (**b**) effect of ${\bar{\lambda }}_{n}$ (from Figure 5), (**c**) effect of ${\bar{\eta }}_{n}$ (from Figure 6), (**d**) effect of ${\bar{\rho }}_{n}$ (**e**) effect of number of layers *n*, and (**f**) effect of ${\bar{\kappa }}_{n}$.

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
