# Peer review of "Start-Up Electroosmotic Flow of Multi-Layer Immiscible Maxwell Fluids in a Slit Microchannel"

_micromachines, 2020, doi:10.3390/mi11080757_

Round 1

Reviewer 1 Report

In this paper, the transient electroosmotic flow of multi-layer immiscible fluids in a slit channel has been investigated. This reviewer highly value the numerical approach, and confirm that there is no error in the equations used in the analysis. 

Although the corresponding experimental results are missing, the results are useful for the microchannel design. 

I recommend this paper to be accepted.

Reviewer 2 Report

In this manuscript, Escandon et al. mathematically modeled the electroosmotic flow of multiple liquids in a microchannel. The maths and physics are solid and well presented. The results look reasonable as well. One thing I suggest the authors to add is the microslit and some basic characterization of it. I was curious how the microchannel looks like. What material is it made from (can affect flow), and how about the Reynolds numbers for the different liquids? A figure with detailed information will be helpful. 
